# Switching of Photocatalytic Tyrosine/Histidine Labeling and Application to Photocatalytic Proximity Labeling

**DOI:** 10.3390/ijms231911622

**Published:** 2022-10-02

**Authors:** Keita Nakane, Haruto Nagasawa, Chizu Fujimura, Eri Koyanagi, Shusuke Tomoshige, Minoru Ishikawa, Shinichi Sato

**Affiliations:** 1Graduate School of Life Sciences, Tohoku University, 2-1-1 Katahira, Aoba-ku, Sendai 980-8577, Japan; 2Frontier Research Institute for Interdisciplinary Sciences, Tohoku University, 2-1-1 Katahira, Aoba-ku, Sendai 980-8577, Japan

**Keywords:** protein chemical labeling, photocatalyst, proximity labeling, tyrosine, histidine

## Abstract

Weak and transient protein interactions are involved in dynamic biological responses and are an important research subject; however, methods to elucidate such interactions are lacking. Proximity labeling is a promising technique for labeling transient ligand–binding proteins and protein–protein interaction partners of analytes via an irreversible covalent bond. Expanding chemical tools for proximity labeling is required to analyze the interactome. We developed several photocatalytic proximity-labeling reactions mediated by two different mechanisms. We found that numerous dye molecules can function as catalysts for protein labeling. We also identified catalysts that selectively modify tyrosine and histidine residues and evaluated their mechanisms. Model experiments using HaloTag were performed to demonstrate photocatalytic proximity labeling. We found that both ATTO465, which catalyzes labeling by a single electron transfer, and BODIPY, which catalyzes labeling by singlet oxygen, catalyze proximity labeling in cells.

## 1. Introduction

Ligand–protein and protein–protein interactions via weak and transient interactions are essential for regulating dynamic biological phenomena. For example, membrane protein G-protein-coupled receptors interact with other proteins in the cytoplasm, cytoskeleton, or extracellular side of the membrane and activation by agonists can dynamically alter the interactome [1]. Identifying these important and dynamic interactions remains challenging. One reason for the difficulty in this analysis may be the weak and transient interactions between ligands and proteins. Conventional methods for identifying proteins that bind to materials include the affinity chromatographic enrichment of binding proteins. However, these methods are only suitable for identifying proteins that bind with high affinity (*K*_D_ < 10^−6^ M) and cannot be used to analyze transient interactions that dissociate during the wash processes of the beads/resins used in affinity chromatography [2]. A typical example of such weak interactions is the binding of sugars to lectins (*K*_D_ > 10^−4^ M) [3], which confers transient binding properties to the cell surface via weak binding and is involved in cell adhesion, differentiation, and cellular signaling, contributing to rapid responses by the cellular system. Tools for analyzing such weak interactions include nuclear magnetic resonance (NMR), isothermal titration calorimetry (ITC), surface plasmon resonance (SPR), and differential scanning calorimetry (DSC). However, these methods are used to detect affinities between one type of protein and one type of ligand and cannot be applied to identify unknown ligand-binding proteins.

Methods for comprehensively identifying ligand-binding proteins independent of affinity strength include protein crosslinking [4], thermal profiling [5], and irreversible covalent labeling of ligand-binding proteins in a proximity-dependent manner. Photoaffinity labeling [6,7,8,9], ligand-directed labeling [10], and catalyst-proximity labeling [11,12] have also been applied in lectin labeling. Among these methods, we focused on catalyst-proximity labeling, which is useful for identifying proteins that interact with analytes independent of their affinity strength. Insoluble membrane proteins that are difficult to analyze using conventional methods can also be evaluated under denaturing conditions. Proximity labeling can detect weak, transient, or hydrophobic protein–protein interactions in their native states, revealing spatial and temporal protein interaction networks that can improve the understanding of specific biological processes.

Proximity labeling has been widely investigated in recent years and is used to analyze protein–protein interactions in living cells [13,14], cell membranes [15,16], ligand recognition on cell membrane surfaces [17,18], and intercellular communication [19]. APEX and BioID are performed to chemically label proteins in proximity to a target protein to which an enzyme is fused [20,21]. Even for weak interactions, proteins that form interactions are labeled in a proximity-dependent manner. Proteins can be enriched by labeling with purifiable tags, such as biotin, and then identified using nanoscale liquid chromatography coupled to tandem mass spectrometry (nanoLC–MS/MS) analysis.

Proximity labeling using small-molecule catalysts has recently been investigated [22,23,24]. We also developed photocatalytic proximity-labeling reactions that proceed in proximity to photocatalysts. In photocatalytic proximity labeling, the timing of labeling can be controlled by visible-light stimulation of the reaction system in an irreversible manner within a few minutes of light exposure. We found that 1-methyl-4-arylurazole (MAUra) is a suitable labeling reagent for photocatalytic proximity labeling [25]. In this reaction, the amino acid residues tyrosine (Tyr) and histidine (His) are labeled; each reaction proceeds via two different mechanisms, as described below (Figure 1). (1) A single-electron transfer (SET) reaction proceeds between MAUra or both MAUra and the Tyr residue and the excited photocatalyst to produce radical species (Figure 1B). (2) The energy-transfer reaction between the excited photocatalyst and dissolved oxygen generates singlet oxygen (^1^O_2_). A covalent bond is formed by a nucleophilic attack of MAUra on electrophilic endoperoxides generated in a reaction between ^1^O_2_ and His residues (Figure 1C, see Appendix A for a description of histidine labeling) [26]. Each reaction is mediated by short-lived active species, radical species, and ^1^O_2_; hence, the reaction occurs in proximity to the catalyst. Reaction (1) is completed at a proximity of approximately 6 nm from the catalyst-linked protein [27], and reaction (2) occurs at approximately 10 nm from the photocatalytic molecule [26]. These labeling radii are suitable for selectively labeling ligand–protein and protein–protein interactions.

However, studies are needed to evaluate the catalyst structures that can catalyze proximity labeling and examine the structure–functionality relationship of catalysts. In this study, we evaluated the effect of catalysts on protein labeling using MAUra as the labeling reagent. In addition, different types of catalysts were used in photocatalytic proximity labeling: each photocatalyst was bound to a HaloTag-fused proteins to determine whether the tagged protein could be selectively labeled in a mixed protein system.

## 2. Results

### 2.1. Photocatalyst Screening

The compounds shown in Figure 2 were screened as candidate photocatalysts. Ru(bpy)_3_Cl_2_ (**2**) [28], fluorescein (**3**) [28], dibromofluorescein (**4**) [29,30,31], rhodamine 123 (**5**) [32], rhodamine B (**6**) [28], BODIPY (**7**) [33], halo-BODIPYs (**8–11**) [34], iodo-coumarin (**12**), eosin Y (**13**) [28], rose bengal (**14**) [28], ATTO465-CO_2_H (**15**) [27], and riboflavin (**16**) [35] were selected as candidate catalysts based on two properties: (a) their ability to be excited by visible light stimulation and (b) their known activities as catalysts for SET or as photosensitizers to produce ^1^O_2_. Each dye also exhibits a different excitation spectrum, and experiments were conducted at a wavelength of either 455 or 540 nm, which is the appropriate light-emitting diode (LED) illumination source, for 5 min on ice.

Ubiquitin was selected as a substrate protein for screening. Ubiquitin contains one Tyr and one His residue, both of which are moderately exposed on the protein surface. Trypsin digestion of ubiquitin generates peptide fragments containing each residue and has a moderate length that is easy to detect (sequences: TLSD*Y*NIQK and ESTL*H*LVLR). We considered ubiquitin as a suitable substrate for evaluating residue selectivity. After labeling, the proteins were trypsin-digested in a sodium dodecyl sulfate–polyacrylamide gel electrophoresis (SDS-PAGE) gel and the resulting peptide fragments were quantitatively compared based on their MS1 intensities determined using nanoLC–MS/MS. We compared the labeling efficiencies of Tyr and His using candidate catalyst molecules based on the efficiency of each residue modified by Ru(bpy)_3_Cl_2_ (**2**). The results are shown in Figure 2C (see Appendix A for MS/MS analysis of labeled peptides).

### 2.2. Mechanistic Analysis of Photocatalysts

We focused on three dyes: Ru(bpy)_3_ complex (**2**), which modifies both Tyr and His; BODIPY (**7**), which is selective for His labeling; and ATTO465 (**15**), which is selective for Tyr labeling.

We evaluated whether SET reactions occurred between the catalysts and MAUra (**1**). The fluorescence of the dye is quenched by adding a substrate. Stern–Volmer fluorescence quenching experiments confirmed that the fluorescence of the dye was quenched by electron transfer between the excited dye and substrate [36]. For BODIPY, we previously showed that adding MAUra does not quench fluorescence, indicating that it does not catalyze SET reactions [33]. The Ru(bpy)_3_ complex underwent SET with MAUra, and the fluorescence of **2** was quenched by the addition of **1** (Appendix A). The SET reaction between **1** and **15**, a selective Tyr labeling catalyst, was evaluated (Figure 3A). The fluorescence of **15** was quenched in a concentration-dependent manner by adding **1**.

The production of ^1^O_2_ by each dye was evaluated using a singlet oxygen sensor green (SOSG), an ^1^O_2_ probe. Comparison of the ^1^O_2_ production efficiency showed results of **7** < **15** < **2**; unexpectedly, **2** and **15** exhibited higher ^1^O_2_ production efficiency compared to **7**, which catalyzed His-selective labeling (Figure 3B and Appendix A).

### 2.3. Photostability Evaluation of Photocatalysts

The stability of the dye structures under photostimulation was evaluated by measuring the fluorescence intensity of each dye after irradiation with 455 nm light for 10 min on ice. The fluorescence intensity of **2** retained 95% of its fluorescence intensity after irradiation, whereas **15** showed extensive fluorescence quenching after irradiation, retaining only 7.7% of its fluorescence. BODIPY (**7**) exhibited a fluorescence intensity of 85%, suggesting that BODIPY is a relatively stable catalyst (Figure 4).

### 2.4. Photocatalytic Proximity Labeling using HaloTag

The proximity dependence of the environment for catalytic protein labeling was evaluated using ligand-conjugated photocatalysts (**17**–**19**) in which each of the three catalysts, Ru(bpy)_3_ complex, BODIPY, or ATTO465, was linked to a HaloTag ligand. HaloTag ligand-conjugated photocatalysts were added to a mixed protein system containing glutathione *S*-transferase (GST)-HaloTag (final concentration: 1 μM) and the cell lysate (1 mg/mL protein). Because BODIPY and ATTO465 exhibit strong fluorescence, we observed their fluorescence after SDS-PAGE, and only bands with molecular weights corresponding to GST-HaloTag were fluorescently labeled (Figure 5, lanes 3 and 7). The results suggested that the ligand-conjugated photocatalysts selectively bound to HaloTag.

Protein mixtures containing each photocatalyst bound to GST-HaloTag were labeled with an azide-linked MAUra derivative (**20**). After the copper-free click reaction of the azide group with dibenzocyclooctyne (DBCO)-Cy5, the labeled proteins were visualized by fluorescence with Cy5. The results are shown in Figure 5 (see Appendix A for the gel image before processing). The estimated target selectivities based on the band intensities of lanes 4, 6, and 8 are shown in Appendix A.

### 2.5. HaloTag-H2B Photocatalytic Proximity Labeling in Cells

Finally, taking advantage of the high membrane permeability of the BODIPY and ATTO465 photocatalysts, we performed intracellular proximity labeling of nuclear histone H2B. When HaloTag-H2B was transiently expressed in HeLa cells, and the cells were treated with HaloTag-ligand-conjugated photocatalysts (**18**, **19**), the photocatalysts showed fluorescence in the nucleus (Appendix A). The subcellular localization of DTB-labeled proteins was visualized with streptavidin-Texas red, and nuclear proteins were selectively labeled (Figure 6).

## 3. Discussion

As shown in Figure 2C, fluorescein (**3**) displayed slight Tyr selectivity but low efficiency. Dibromofluorescein (**4**), a promising photosensitizer, exhibits good His selectivity. The mitochondria-localized fluorescein derivative **5** showed similar residue selectivity and efficiency to **2**. Rhodamine B (**6**) exhibited a low Tyr labeling efficiency and high selectivity for His.

We previously reported that BODIPY (**7**) is a suitable catalyst for His labeling when a peptide is used as the substrate [33]. Figure 2C illustrates that the His-labeling efficiency of **7** was 4-fold higher than that of **2**, even when ubiquitin was used as a substrate. The Tyr labeling efficiency was low and showed high His selectivity. Although introducing halogen atoms into BODIPY increases the efficiency of ^1^O_2_ production [28,34,37,38,39,40], our previous experiments suggested that MAUra is degraded when ^1^O_2_ is overproduced. When ubiquitin was used as a substrate, halo-BODIPYs **8**–**11** showed a similar or lower His labeling efficiency compared to that of BODIPY **7**. I-Coumarin (**12**) was also evaluated and expected to show similarly elevated ^1^O_2_ production; however, the reaction efficiency was lower than that of **7**. Tyr labeling was not observed, indicating that **12** was a catalyst with high His selectivity.

Eosin Y (**13**) [41] and rose bengal (**14**) [42] exhibit photoredox properties that catalyze SET and may be useful catalysts for Tyr labeling. The Tyr labeling efficiency of these two dyes was approximately 3-fold higher than that of **2**, and high Tyr selectivity was observed. ATTO465-CO_2_H (**15**, an acriflavine derivative), which also catalyzes SET [43], exhibited a higher Tyr labeling efficiency compared to the other catalysts. Riboflavin (**16**) has also been reported to catalyze SET; although we expected it to be useful as a catalyst for Tyr labeling, riboflavin showed low catalytic activity in our experimental system.

Figure 3A and Appendix A show that the fluorescence of **2** and **15** is quenched in the presence of **1**. These results suggest that the catalysts lead to Tyr labeling via single-electron oxidation of MAUra to produce MAUra radical species. MAUra can be oxidized at an electrical potential of 0.6 V [25,44], whereas the oxidizing potentials of the Ru(bpy)_3_ complex and ATTO465 flavin skeleton are 1.27 [45] and 1.48 V [28], respectively. These reported oxidation potentials are consistent with the results of our Stern–Volmer fluorescence quenching experiments. We tested the reactivity of the MAUra radical species with redox-active residues other than Tyr under single-electron oxidation conditions (Appendix A). Using **1** and **15**, methionine (Met) residues were oxidized but not labeled. Oxidation and labeling of tryptophan (Trp) residues were also observed, although these effects were less pronounced than the labeling of Tyr. As ubiquitin does not contain Trp, which is present in low abundance and rarely found on protein surfaces, Tyr is likely the main target of the radical reaction using MAUra. Efficient labeling of Trp at specific sites may be possible by placing the catalyst in proximity to Trp.

Figure 2C, Figure 3 and Appendix A show that **2** and **15** induce the single-electron oxidation of **1**, and that **2**, **15**, and **7** can produce ^1^O_2_. In addition, **7** and **15** catalyzed His-selective labeling and Tyr-selective labeling, respectively. These results suggest that when SET does not occur, and MAUra is not oxidized, the pathway through which nucleophilic MAUra labels His, oxidized by ^1^O_2_, is preferential (Figure 1C). In contrast, when MAUra was radicalized, the Tyr labeling pathway proceeded preferentially, even in the presence of ^1^O_2_ (Figure 1B). A dye causing SET may also quench its function as a photosensitizer in the presence of MAUra by SET reactions, thus suppressing ^1^O_2_ production and His labeling and helping Tyr labeling to proceed preferentially. The difference in residue selectivity presented in Figure 2C suggests that these two pathways are controlled by the ratio of the contributions of SET and ^1^O_2_ production. In site-selective functionalization of proteins with multiple reaction sites rather than simple substrates, such as ubiquitin, it is necessary to selectively bind the photocatalyst to the protein and control proximity labeling. For example, we achieved Fc region-selective functionalization by placing a photocatalyst in proximity to the Fc region of an antibody [26,33].

BODIPY and ATTO465 show strong fluorescence and can be used to determine where the catalyst is bound in cells prior to labeling. In the case of covalent ligands, such as HaloTag, selective binding to proteins of interest can be confirmed in experiments such as those shown in Figure 5 and lanes 3 and 7.

Figure 4 shows that ATTO465 was degraded after a few minutes of light exposure, whereas BODIPY fluorescence of the HaloTag was still observed after light irradiation. These results suggested that BODIPY can be used as a reporter to observe photocatalyst localization even after labeling. ATTO465 is spontaneously deactivated within a short time after photo-irradiation without inducing excessive labeling reactions. Therefore, ATTO465 is considered suitable for controlling proximity labeling with high time resolution.

Figure 5 shows that GST-HaloTag was selectively modified in the Ru(bpy)_3_ complex with satisfactory reaction efficiency. This result can be attributed to the high photostability of the Ru(bpy)_3_ complex, making it an excellent catalyst, and the efficient progression of Tyr and His labeling in the proximity space of the catalyst. In addition, His labeling selectively proceeded in GST-HaloTag with BODIPY, although with lower efficiency. In contrast, when ATTO465 was used, selectivity was low, possibly because of the nonspecific interaction properties of ATTO465. This feature may be disadvantageous in applications for target identification using ligand-conjugated ATTO465, as it yields a high background reaction. In contrast, ATTO465 appropriately catalyzed proximity labeling in the intracellular environment (Figure 6) [27]. The nonspecific adsorptive property of ATTO465 may have enabled labeling of the protein surrounding tagged proteins without strong adverse effects when ATTO465 was introduced on HaloTag in the cells.

We previously showed that ATTO465 is an effective photocatalyst in cells [27]. In this study, we showed that various other fluorescent molecules also produce radical species and ^1^O_2_ to label proteins. In addition, the mechanism of protein modification by each fluorescent molecule differed. We also showed that BODIPY catalyzes proximity labeling in cells (Figure 6).

## 4. Materials and Methods

### 4.1. Materials

Ubiquitin was purchased from R&D Systems (Minneapolis, MN, USA). The GST-HaloTag and HaloTag-H2B plasmids were purchased from Promega (Madison, WI, USA). Compounds **2**, **13**, and **14** were purchased from Tokyo Chemical Industry (Tokyo, Japan); compounds **3** and **6** were purchased from Nacalai Tesque (Kyoto, Japan); and compounds **4**, **5**, **7**, **15**, and dibenzocyclooctyne-Cy5 were purchased from Sigma-Aldrich (St. Louis, MO, USA). All reagents were used without further purification. Compounds **8**–**10** and **12** were prepared as previously reported [33,46]. Compounds **1** and **20** were synthesized as previously described [25]. HaloTag-ligand-conjugated photocatalysts **17**–**19** were synthesized as described previously [27,33].

### 4.2. Ubiquitin Labeling

The photocatalyst (from a 100 mM stock solution in dimethyl sulfoxide (DMSO), final concentration 1 mM) was added to ubiquitin (10 µM) in a 10 mM MES buffer (pH 7.4) (50 µL). MAUra (**1**) (100 mM in DMSO, final concentration 500 µM) was then added to the solution. The solution was irradiated with blue light (RELYON, Tokyo, Japan, Twin LED light, 455 or 540 nm) for 5 min on ice. The reaction mixture was added to 5× sample buffer (final concentration 50 mM Tris-HCl pH 6.8, 2% SDS, 0.025% bromophenol blue, 10% glycerol), heated at 95 °C for 5 min, and separated by SDS-PAGE using 4–20% acrylamide gels (Bio-Rad, Hercules, CA, USA).

### 4.3. In-Gel Digestion of Labeled Ubiquitin

According to Section 4.2, ubiquitin (10 µM) was labeled with **1**. The labeled proteins were separated using SDS-PAGE. Bands corresponding to labeled ubiquitin were separated, and the excised bands were cut (approximately 1 mm pieces). Gel pieces were transferred into microtubes, and 1 mL of water was added and incubated at 37 °C for 10 min. The solution was removed, and the washing procedure was repeated three times. For destaining, 50% CH_3_CN in 100 mM Tris buffer (pH 8.0) was added and incubated at 37 °C for 10 min, and the solution was removed. CH_3_CN was added to the tubes for dehydration and then incubated at 37 °C for 10 min. After the solution was removed, trypsin solution was added to each tube and incubated overnight at 37 °C. The reaction was quenched by adding aqueous trifluoroacetic acid solution (final concentration 0.1% *v*/*v*) and desalted using C18 pipette tips (Nikkyo Technos Co., Ltd., Tokyo, Japan). After desalting, the solvent was removed by centrifugation.

### 4.4. NanoLC-MS/MS Analysis

NanoLC-MS/MS analysis was performed using LC-nano-ESI-MS comprising a quadrupole time-of-flight mass spectrometer (Triple TOF^®^ 5600 system; SCIEX, Framingham, MA, USA) equipped with a nanospray ion source and nanoLC system (Eksigent Nano LC Ultra 1D Plus; SCIEX). The trap column used for nanoLC was a NanoLC Trap ChromXP C18, 3 μm 120 Å (SCIEX), and the separation column was a 12.5 cm × 75 μm capillary column packed with 3 μm C18-silica particles (Nikkyo Technos Co., Ltd., Tokyo, Japan). The micropump (flow rate 300 nL/min) gradient method was used as follows: mobile phase A: 2% acetonitrile, 0.1% formic acid; mobile phase B: 80% acetonitrile, 0.1% formic acid aq. 0–20 min: 5–45% B, 20−21 min: 45–100% B, 21–26 min: 100% B. NanoLC-MS/MS data were acquired in an information-dependent acquisition mode controlled by Analyst^®^ TF 1.5.1 software (SCIEX). The settings for the data-dependent acquisition were as follows: accumulation time, 0.25 s; full MS (MS1, TOF-MS) scan range, 400–1250 *m*/*z*, excluding the former target ion for 12 s; and mass tolerance, 50 mDa. The top 10 signals were selected from MS2 scanning per full MS scan. The MS2 (product ion) scan accumulation time and range were 0.05 s and 100–1500 *m*/*z*, respectively. All experiments were conducted in triplicate. MS/MS spectra were searched against the respective amino acid sequence (ubiquitin) using MaxQuant (Freeware) [47] with default settings. A FASTA file corresponding to the ubiquitin sequence was used. For labeling, oxidation (+O) of His, Met, and Tyr residues, acetylation (+C_2_H_2_O) at the N-terminus, an adduct of MAUra (+C_9_H_7_N_3_O_2_; +189.054 Da) for Tyr residues, and an adduct of MAUra (+C_9_H_7_N_3_O_3_; +205.049 Da) for His residues were set as possible modifications.

### 4.5. Measurement of Singlet Oxygen Generation

The fluorescence intensity of SOSG (Thermo Fisher Scientific, Waltham, MA, USA) oxidized by ^1^O_2_ was determined from the fluorescence of the HPLC peaks. The photocatalyst (2 µM) and SOSG (10 µM) were added to 50% CH_3_CN solution in 10 mM MES buffer (pH 7.4) in a 1.5 mL tube. The solution was irradiated with blue light (RELYON, Twin LED light, 455 nm) for 30 s on ice. After irradiation, the solution was diluted 2.6-fold with 0.1% aqueous formic acid and analyzed using HPLC. Analytical HPLC was carried out on a JASCO PU-4580 HPLC Pump, JASCO LG-4580 Quaternary Gradient Unit (Tokyo, Japan), and JASCO DG-4580 Degassing Unit with a JASCO MD-2018 Plus Photodiode Array Detector, JASCO CO-4060 Column Oven, JASCO As-455 HPLC Autosampler, and JASCO LC-NetII/ADC Interface Box using a C18 reverse phase column (Inertsil ODS-4, 150 × 4.6 mm, 5 μm (GL Science, Inc., Tokyo, Japan)). The HPLC conditions were as follows: mobile phase A, 0.1% formic acid in H_2_O, mobile phase B, 0.1% formic acid in CH_3_CN. 0−5 min, 5% B; 5−27 min, 5−100% B; 27−32 min, 100% B. The fluorescence of the separated peaks was detected using HPLC (Ex 504 nm/Em 525 nm).

### 4.6. Stern–Volmer Fluorescence Quenching Experiments

Fluorescence spectra were measured using a JASCO FP-6500 instrument after mixing the photocatalyst and **1**. For **2**, we used 50 nM **2** in 45% CH_3_CN and 50% DMSO in 10 mM MES buffer (pH 7.4) with 0 or 50 mM **1**, at an excitation wavelength of 455 nm. For **15**, we used 10 nM **15** in 10% DMSO in 10 mM MES buffer (pH 7.4)) with 0, 10, 50, or 100 mM **1**, with an excitation wavelength of 463 nm.

### 4.7. Evaluation of Photostability

The photocatalyst (500 µM) was added to 50 μL of 10 mM MES buffer (pH 7.4; 50 μL). The solution was irradiated with blue light (RELYON, Twin LED light, 455 nm) for 10 min on ice. After irradiation, the solution was diluted 2.6-fold with 0.1% aqueous formic acid and analyzed using HPLC. A micropump (1 mL/min) gradient method was used as follows: mobile phase A: 0.1% formic acid; mobile phase B: 100% CH_3_CN. 0−5 min: 5% B, 5−27 min: 5−100% B, 27−32 min: 100% B. The absorbance of the separated peaks was detected using HPLC (**2**: 450 nm, **7**: 500 nm, **15**: 450 nm).

### 4.8. GST-HaloTag Labeling in the Protein Mixture

Dye-conjugated HaloTag ligand (final concentration, 1 µM) was added to a protein mixture of GST-HaloTag (1 µM) and HEK293FT cell lysate (1 mg/mL) in 10 mM MES buffer (pH 7.4) (50 µL). The solution was incubated at 37 °C for 1 h. Labeling reagent **20** (from 100 mM solution in DMSO, final concentration 500 µM) was added to the mixture. The solution was irradiated with blue light (RELYON, Twin LED light, 455 nm) for 5 min on ice. The reaction mixture was added to 2-iodoacetamide (from 100 mM solution in H_2_O, final concentration 2 mM) and incubated at 37 °C for 1 h. Dibenzocyclooctyne (DBCO)-Cy5 (final concentration 500 µM) was added to the solution of azide-conjugated GST-HaloTag and incubated for 1 h at 37 °C. The reaction mixture was added to 5× sample buffer (final concentration 50 mM, pH 6.8, 2% SDS, 0.025% bromophenol blue, 10% glycerol), heated at 95 °C for 5 min, and then separated using SDS-PAGE on 4–20% acrylamide gels (Bio-Rad). The fluorescence of the labeled proteins was detected using a Molecular Imager Fusion Solo S (VILBER LOURMAT, Collégian, Paris, France). After obtaining fluorescent images, the same gel was visualized using Coomassie brilliant blue (CBB) staining.

### 4.9. Peptide Labeling

ATTO465-CO_2_H (**15**) (from a 100 mM stock solution in DMSO, final concentration 100 µM) was added to a solution of peptides (100 µM) in 10 mM MES buffer (pH 7.4) (50 µL). MAUra (**1**) (100 mM in DMSO, final concentration 200 µM) was added to the solution and then irradiated with blue light (RELYON, twin LED light, 455 nm) for 5 min on ice. The reaction mixture was diluted 50-fold with 0.1% trifluoroacetic acid and mixed with CHCA solution (5.0 mg/mL solution in acetonitrile: 0.1% trifluoroacetic acid aq. = 0.5 µL:0.5 µL). The mixture was placed on a MALDI-TOF plate and dried at room temperature. The modified protein peaks were detected using MALDI-TOF MS analysis (ABSCIEX TOF/TOF^TM^ 5800).

### 4.10. Transfection of HaloTag-H2B

HaloTag-H2B was transfected according to the protocol of the Avalanche^®^Omni Transfection Reagent (APRO Science, Tokushima, Japan). HeLa cells were seeded at a density of 4 × 10^5^ cells/mL in 1.5 mL media into lysine-coated dishes and incubated for 24 h in a CO_2_ incubator.

### 4.11. HaloTag-H2B Proximity Labeling in Cells

HaloTag-H2B-transfected HeLa cells were incubated in a CO_2_ incubator for 24 h. The cells were treated with a ligand-conjugated photocatalyst (from a 10 mM stock solution in dimethylformamide; final concentration 5 μM in medium) and incubated in a CO_2_ incubator for 2 h. After removing the medium, HaloTag-H2B-transfected HeLa cells were washed twice with 2 mL of phosphate-buffered saline (PBS) to remove unbound photocatalysts. After washing, 500 µL of 500 µM MAUra-DTB (**21**) solution in PBS was added to the dish. The cells were incubated at room temperature for 30 min and then photoirradiated on ice (455 nm, 5 min for **18**; 1 min for **19**). After the labeling reaction, the cells were gently washed once in a dish with 2 mL of PBS and fixed with 4% paraformaldehyde for 30 min at room temperature. After removing the paraformaldehyde solution, the dish was washed once with 2 mL of PBS and treated with 0.4% Triton X for 5 min at room temperature. The cells were blocked for 5 min at room temperature with a blocking solution (ImmunoBlock, KAC, Hyogo, Japan) and washed once with 2 mL PBS. The dish was incubated with streptavidin, TEXAS RED^®^ Conjugate (Sigma-Aldrich) solution diluted 200-fold in TBS-T (20 mM Tris, 150 mM NaCl, and 0.1% Tween 20, pH 7.6), and incubated for 12 h at 4 °C. The cells were washed twice with 2 mL of PBS, and nuclei were counterstained with Hoechst33342. Fluorescence signals were observed using a confocal laser microscope (LMS710 Spectral Confocal System; Zeiss, Oberkochen, Germany). The results are shown in Figure 6 and Appendix A.

## 5. Conclusions

By examining various fluorescent dyes as photocatalysts for protein labeling, we found that many dye molecules can function as photocatalysts for protein labeling. The catalytic ability for SET and ^1^O_2_ production differed depending on the function of the dyes. Accordingly, the labeling efficiencies of Tyr and His differed. ATTO465 was the preferred catalyst for Tyr selective labeling, and BODIPY was the preferred catalyst for His selective labeling.

We also evaluated the target selectivity of proximity labeling in complex protein mixtures using HaloTag ligand-conjugated catalysts and GST-HaloTag. The Ru(bpy)_3_ complex and BODIPY selectively labeled GST-HaloTag with less nonspecific adsorption. In contrast, ATTO465 induced nonspecific protein labeling, possibly because of nonspecific adsorption. Furthermore, photocatalysts bound to HaloTag catalyzed proximity labeling using intracellularly expressed HaloTag-H2B, as demonstrated in labeling with ATTO465 and BODIPY.

These findings revealed a new functional aspect of molecules used as fluorescent dyes in photocatalysts for proximity labeling. Such catalytic fluorescent dyes are not limited to organometallic complexes, such as ruthenium and iridium complexes, thus expanding the application of photocatalytic proximity labeling. These molecules, which can easily cross cell membranes, can help control intracellular reactions, which is a current challenge in proximity labeling using artificial molecules. Reagents for introducing photocatalyst moieties, such as *N*-hydroxysuccinimide (NHS) esters of each dye molecule, are commercially available, and it is easy to functionalize small molecules, peptides, and biomolecular ligands using these dyes. We are currently evaluating proximity labeling to identify weak and transient binding proteins using photocatalyst-conjugated bioactive molecules.

## Figures and Tables

**Figure 1 ijms-23-11622-f001:**
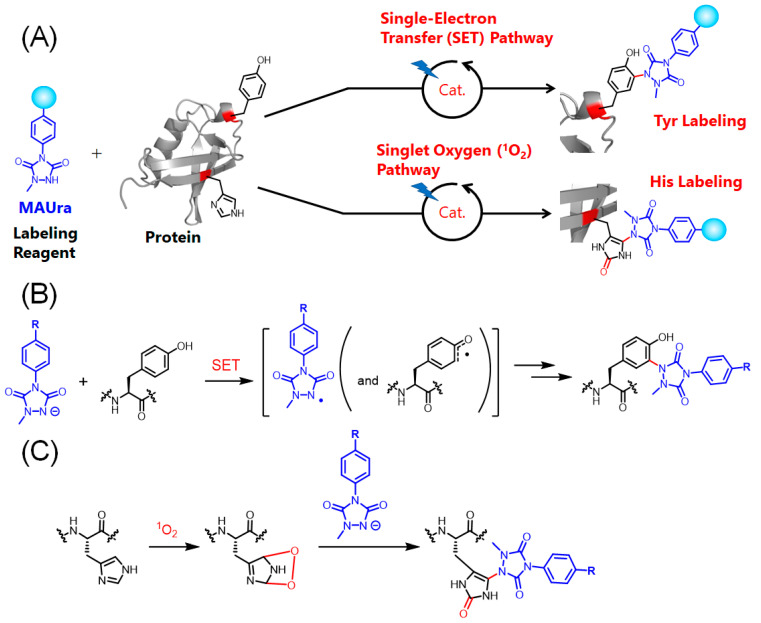
Proximity labeling of the photocatalyst. (**A**) 1-methyl-4-arylurazole (MAUra) labels tyrosine (Tyr) and histidine (His). (**B**) Proposed mechanism of Tyr labeling. (**C**) Proposed mechanism of His labeling.

**Figure 2 ijms-23-11622-f002:**
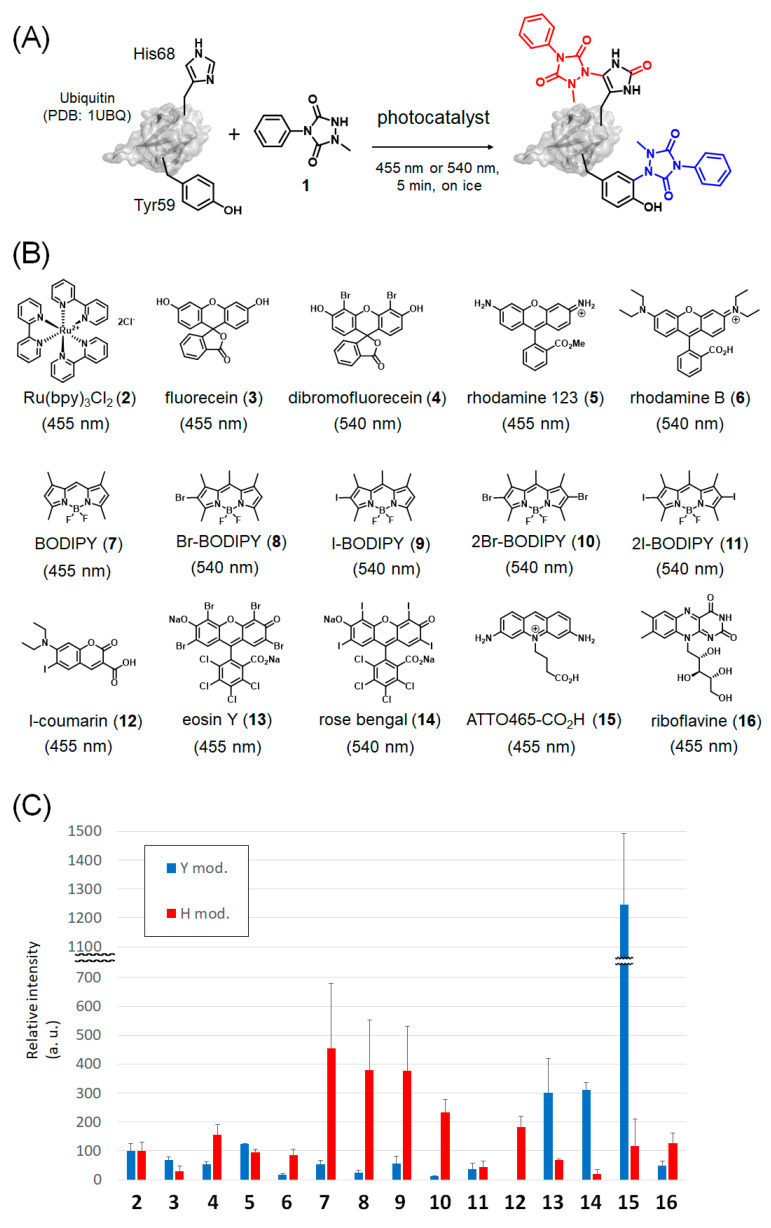
Screening of photocatalysts. (**A**) Scheme of ubiquitin labeling. (**B**) Chemical structures of photocatalyst candidates. (**C**) Relative MS1 intensity of labeled peptide fragments.

**Figure 3 ijms-23-11622-f003:**
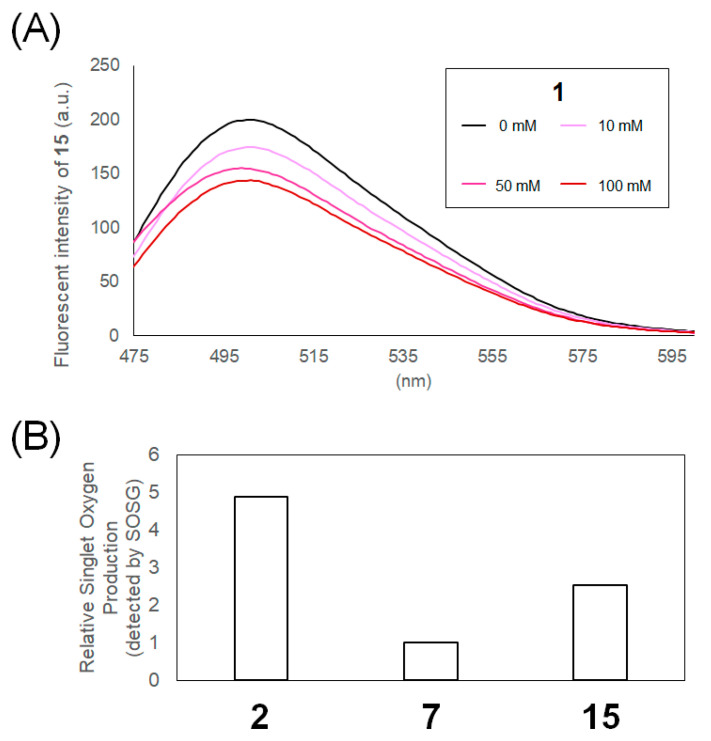
Single-electron transfer (SET) and ^1^O_2_ production properties of photocatalysts. (**A**) Stern–Volmer fluorescence quenching experiment. Compound **15** (10 nM, in 10% DMSO in 10 mM MES buffer (pH 7.4)) with 0, 10, 50, or 100 mM **1**. (**B**) Relative singlet oxygen production by compounds **2**, **7**, and **15**. See Appendix A for high-performance liquid chromatography (HPLC) data to detect the oxidized singlet oxygen sensor green (SOSG).

**Figure 4 ijms-23-11622-f004:**
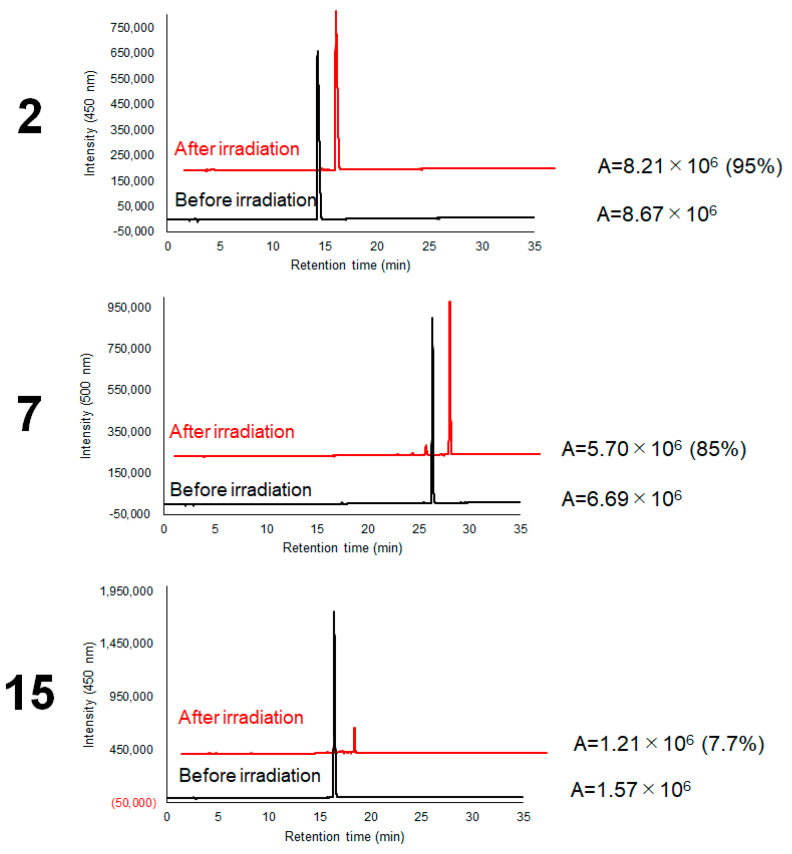
Photostability of **2**, **7**, and **15**.

**Figure 5 ijms-23-11622-f005:**
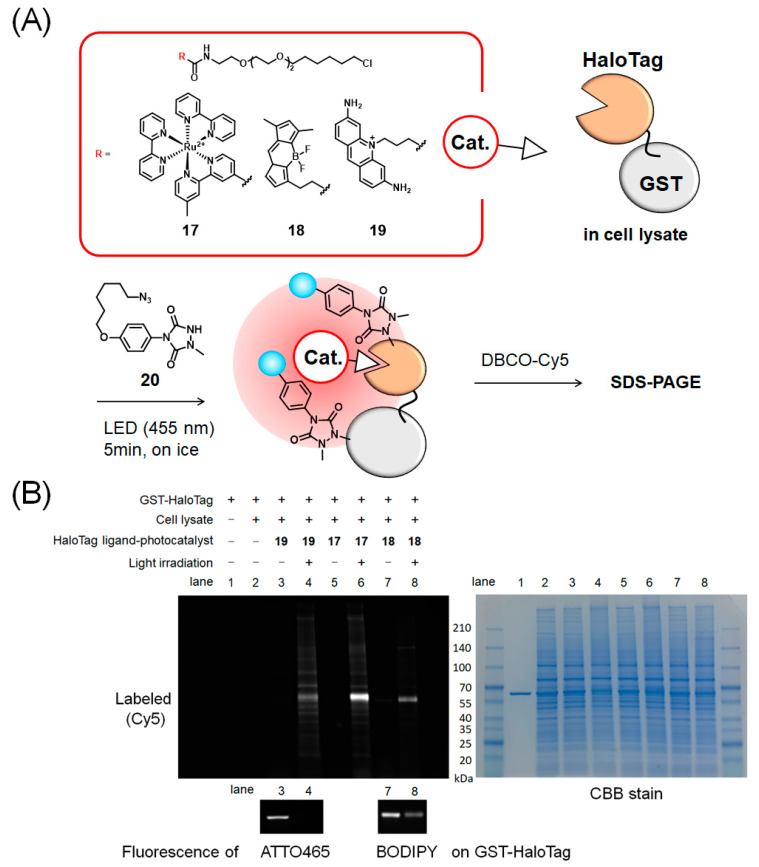
Photocatalytic proximity labeling using HaloTag ligand-conjugated photocatalysts and gluatione S-transferase (GST)-HaloTag. (**A**) Scheme of the labeling. (**B**) Detection of labeling and fluorescence of photocatalyst on GST-HaloTag.

**Figure 6 ijms-23-11622-f006:**
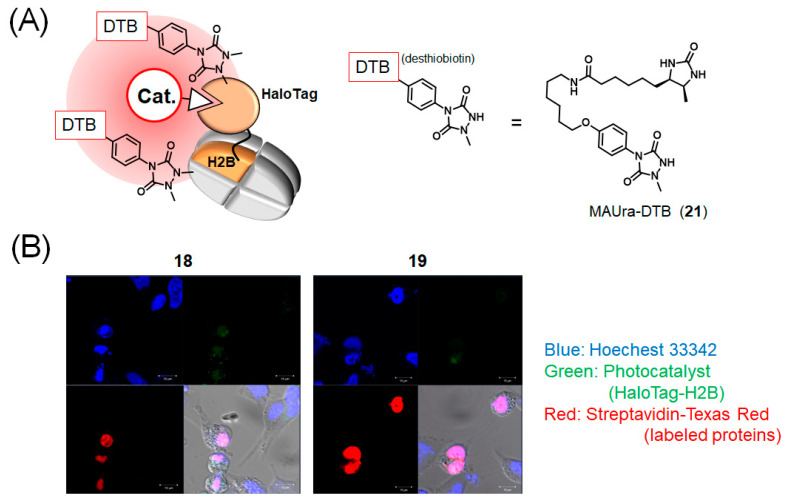
HaloTag-H2B photocatalytic proximity labeling in cells. (**A**) Scheme of the labeling and structure of MAUra-DTB (**21**). (**B**) Detection of labeled protein localization and fluorescence of photocatalyst on HaloTag-H2B. Top left: Hoechest 33342. Top left: Photocatalyst. Bottom right: streptavidin-Texas red. Bottom right: Overlay of bright-field image and three-color fluorescent image.

## Data Availability

Data are present within the article or supplementary material.

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
