# Peer review of "Switching of Photocatalytic Tyrosine/Histidine Labeling and Application to Photocatalytic Proximity Labeling"

_ijms, 2022, doi:10.3390/ijms231911622_

Round 1

Reviewer 1 Report

In this paper, the authors reported Tyr/His-selective proximity labelling method using 1-methyl-4-arylurazole (MAUra) as an anchoring molecule, and several kinds of photocatalysts were compared in the manuscript. As a result, ATTO465 was found suitable for Tyr, and BODIPY was found suitable for His. The results reported in the manuscript support this conclusion well, but this paper is judged to be of sufficient for publication after minor revisions.

Comments:

1. During His modification, MAUra seems to be able to attack two kinds of carbon atom within the endoperoxide of His(O2) intermediate. There are two kinds of carbon atoms adjacent to the dioxide moiety. Does this labelling reaction selectively afford the isomer written in the manuscript?

2. If MAUra radical is involved in the modification reaction, could MAUra radical be attached to e.g. Met or Trp? Please discuss this point in the manuscript.

3. “adduct of MAUra (+C9H6N2O2; +174.043 Da) for Tyr residues, and adduct of MAUra (+C9H6N2O3; + 190.038 Da) for His residues were set as possible modifications.” is written in the manuscript. Is it correct? +C9H7N3O2 may be correct for Tyr modification, and +C9H7N3O3 may be correct for Tyr modification. Please re-confirm this point.

Reviewer 2 Report

These findings reveal a new functional aspect of molecules that have been used as fluorescent dyes as photocatalysts for proximity labeling. I think this work is interesting but the author should try more experiments such utlizing this method in cell experiments not just in cell lysate since these compounds have good cell membrane permeability. Another concen is that since MAUra was reported by author themself in CC, the author should discussed the different between this work and CC.
354

Author Response

Thank you for reviewing this manuscript and for your helpful comments. We will not be able to meet the 5-day limit, but we are currently working on cell-based experiments and would be happy to respond to your comments on Reviewer 2 at a later date.

Reviewer 3 Report

The writing style and grammar need attention. Especially the Abstract needs refining. Briefly explain interactome. Proteins can be labeled, but “interactions” not. Correct the language accordingly.

Introduction needs more specifics and reference support. Give examples of dynamic biological phenomenon.

Line 29. Give at least one more example to the “unconventional methods”.

Line 46. Explain why the authors focused on catalyst-proximity labeling.

Figure 1. Legend. Add “proposed” in front of “mechanism” for both (B) and (C).

It appears that this methodology may be suitable primarily for ubiquitin but not others proteins, especially those possessing multiple, similarly reactive, amino acids. This would be a bigger problem if these amino acid residues are localized within close proximity. Authors should address this potential challenge and convince the readers for how their methodology can be generalized.
